# Examining Reciprocal Links between Parental Autonomy-Support and Children’s Peer Preference in Mainland China

**DOI:** 10.3390/children8060508

**Published:** 2021-06-16

**Authors:** Na Hu, Muzi Yuan, Junsheng Liu, Robert J. Coplan, Ying Zhou

**Affiliations:** 1Shanghai Key Laboratory of Mental Health and Psychological Crisis Intervention, School of Psychology and Cognitive Science, East China Normal University, 3663 Zhongshan Road (N.), Shanghai 200062, China; huna54@163.com (N.H.); ymzkate@outlook.com (M.Y.); 2Department of Psychology, Carleton University, Ottawa, ON K1S 5B6, Canada; robertcoplan@cunet.carleton.ca; 3China Executive Leadership Academy Pudong, 99 Qiancheng Road, Shanghai 201204, China

**Keywords:** parental autonomy-support, peer preference, Chinese children

## Abstract

The present study examined the longitudinal relations between child perceptions of parental autonomy-support and peer preference in mainland China. Participants were *N* = 758 children (50.8% boys; *M_age_* = 10.78 years, *SD* = 1.03 at Wave 1; *M_age_* = 11.72 years, *SD* = 1.11 at Wave 2; *M_age_* = 12.65 years, *SD* = 0.95 at Wave 3) from elementary and middle schools in Shanghai, P.R. China. Children were followed over three years from Grades 4–6 to Grades 6–8. Each year, children reported their perceived maternal/paternal autonomy-support and peer preference (being well-liked among peers) was measured via peer nominations. Among the results, peer preference positively predicted later perceptions of maternal and paternal autonomy-supportive parenting, whereas autonomy-supportive parenting did not significantly predict later peer preference. Results are discussed in terms of the interactions between parental autonomy-supportive parenting and children’s peer relationships in Chinese culture.

## 1. Introduction

Drawn from the conceptual framework of Self-Determination Theory (SDT) [1], autonomy-support refers to parental practices that allow children to make appropriate choices and express themselves, acknowledge children’s feelings, provide explanations and reasons for rules, and afford prompt responses and guidance when children are in need [2,3]. Further, autonomy-supportive parents are motivated to help children find meaning and develop internal interest in activities (e.g., schoolwork, housework) and to treat others with kindness [4,5]. In this regard, autonomy-supportive parenting can be considered as the antithesis of psychological control, a parenting practice that espouses guilt induction as a way to pressure children into thinking or behaving in ways that are in accordance with parental wishes [6,7]. It has also been suggested that parental autonomy-support (or autonomy-granting) should be considered as part of the constellation of positive parenting practices that includes responsiveness, warmth, and involvement [8,9].

Overall, autonomy-supportive parenting practices is theorized to promote the development of children’s perceived independence, volition, and self-sufficiency, characteristics which, in turn, enhance social functioning and the establishment of positive relationships [10,11]. There is some previous empirical evidence of concurrent associations between parental autonomy-support and children’s positive peer relationships in Western societies [12,13,14]. However, parenting and its impact on children’s social competencies can vary across cultural contexts [15]. For example, authoritarian parenting is characterized by demandingness combined with lower responsiveness, which also reflects low levels of parental autonomy-support [16]. Some cross-cultural research reveals that, in contrast to findings with European-American families, authoritarian parenting can impart some benefits among Chinese-American [17] and African-American families [18]. There is also some evidence to suggest that Chinese-American and African-American children are more likely to perceive that their authoritarian family cares for and protects them, as compared to European-American children [16]. However, this evidence reflects Asian families in the Western context. 

Therefore, it is of continued interest to further explore the impact of parental autonomy-support on child social competencies in Asian cultural contexts such as mainland China. To date, results from the few studies conducted among Asian samples have been somewhat mixed [19,20,21], which raises questions about the generalizability of these effects to other cultural contexts. Moreover, most previous studies have been cross-sectional in design, which does not allow for clarification of the directions of effects. Thus, the main goal of the current study was to examine the longitudinal relations between maternal and paternal autonomy-support and peer preference (i.e., being well-liked among peers) among Chinese children. 

### 1.1. Parental Effects: Autonomy-Supportive Parenting → Peer Preference

Autonomy-supportive parenting may facilitate children’s peer preference via several underlying conceptual mechanisms. For example, according to SDT [3], this type of parenting promotes three inherent psychological needs of children: (1) autonomy (feeling ownership of one’s performance); (2) competence (feeling a sense of control); and (3) relatedness (feeling allied with others). Autonomy-supportive parenting contributes to fulfilling these basic psychological needs, which in turn is conducive to positive outcomes, including in the areas of social and psychological adjustment [10]. Parents who espouse the practices of autonomy-support also tend to accept their children’s views, which may help to promote positive self-evaluations and general satisfaction among their children. In the peer context, children with these characteristics are more likely to enjoy meaningful social activities, which in turn facilitates the formation and maintenance of positive peer relationships [11,22]. In contrast, children of mothers who reported lower levels of autonomy-support tend to be less well-accepted by peers [23]. 

There is accumulating evidence of the positive impact of autonomy-supportive parenting on peer relations in late childhood and adolescence. For example, Updegraff et al. [24] found that paternal autonomy-support in open communications with sons was positively related to the level of intimacy in adolescent friendships. Similarly, Williams and colleagues [25] reported that American high school students who perceived a lower level of autonomy-supportive parenting were less likely to experience meaningful peer relationships. Thus, overall, autonomy-supportive parenting is expected to contribute to children’s positive peer relationships. However, to our knowledge, there has been no previous longitudinal research examining *bidirectional* links between autonomy-supportive parenting and aspects of peer relationships. Accordingly, we tested these relations over time, in order to explore possible causal mechanisms that may underlie these associations.

There is also growing evidence to suggest that the meaning and functions of autonomy-support may vary across cultures [26,27]. For example, Asian cultures tend to put greater emphasis on interpersonal harmony and interdependence, whereas individuality and autonomy are prioritized in Western cultures [28]. Accordingly, in Asian cultures, an interdependence-orientation promotes the use of parental control (rather than autonomy-support) to help socialize children to “fit in” [29]. As a consequence, autonomy-support may not have the same positive effects on children’s social functioning in Asian cultures as in the West. There is at least some previous preliminary empirical support for this assertion. Liu and colleagues [30] reported no significant association between maternal autonomy-support and indices of child social functioning in a mainland Chinese sample. As noted previously, other findings from Asian samples have been somewhat mixed [19,20,21]. For example, Soenens et al. [31] found that autonomy-support promoted psychological benefits in a South Korean sample, and effects related to controlling parenting were mediated by the sense of collectivism.

Of note, over the last 20 years, China has undergone socialist marketing economic reforms, which appear to have contributed to large-scale societal changes, including the incremental acceptance and encouragement of autonomy, assertiveness, and competitiveness [32]. As a consequence, we speculated that autonomy-support may now contribute positively to child social functioning in contemporary Chinese society, particularly in urban areas that have experienced the greatest changes [33]. There appears to be some recent empirical support for this notion. Li et al. [13] found that children with parents who espouse autonomy-supportive and warm parenting practices prefer to focus on cooperation and harmony during social interaction with peers and are more accepted by peers. As well, in a recent short-term longitudinal study, Gao et al. [34] reported that maternal autonomy-supportive parenting was positively linked with sociability one year later among girls. Taken together, it appears that autonomy-supportive parenting now facilitates positive social interactions and relationships with peers in contemporary urban China. 

### 1.2. Child Effects: Peer Preference → Autonomy-Supportive Parenting 

Aspects of children’s peer relationships may also impact autonomy-supportive parenting. Attachment theory posits that in childhood, family and peers act as two intertwined “energy” systems, but usually flowing from parents to peers [35]. However, it has also been argued that because of prominent changes that occur in the organization of the attachment system in late childhood and adolescence, the effects of peers may come to overshadow parental influences [36]. In this regard, peer relations can also directly influence parenting practices [37]. According to ecological system theory, family and peer contexts are discrepant and interdependent micro-systems that interact with each other over time [38]. In the peer context, children develop and construct adaptive relationships through favorable social interactions, which facilitates well-being [39]. This may be particularly notable in adolescence, when increasingly more time is spent with peers and their influence is increased as compared to parents [40]. Adolescence is also a transitional period that exacerbates psychosocial vulnerability [41,42], and during which drug use, antisocial behaviors, and other difficulties increase [43]. Therefore, establishing positive and healthy peer relationships are essential to help adolescents to achieve autonomy, maturity, and social skills. In this regard, when parents receive positive feedback from their children, they will be more likely to grant them more autonomy rights. Given these transactional and mutual interrelations, it is plausible that peer relationship may also positively affect autonomy-supportive parenting practices.

There are some empirical findings in support of such child-driven effects [44,45]. For example, Feldman and Wentzel [44] found that early adolescents’ peer liking was positively linked with social support from family. In the same vein, Barbot et al. [45] reported that child social competence positively predicted parental involvement and appropriate monitoring over a five-year period. Finally, Owens and colleagues [46] found that children’s positive interactions with peers predicted positive parenting styles. Taken together, this evidence suggests that peer preference might also contribute to parental autonomy-support.

### 1.3. Role of Gender

We also investigated potential gender differences in the longitudinal associations between autonomy-support and peer preference. Even though SDT suggests that autonomy- support is beneficial for both boys and girls [3], some gender differences have been reported in terms of parenting effects. For example, the positive association between maternal supportive parenting and children’s emotion regulation was found to be more prominent for girls than for boys [47]. Similarly, Liu and colleagues [30] reported that the association between maternal promotion of relatedness and child leadership among peers was stronger for girls than for boys. As mentioned previously, there is some evidence to suggest that boys’ friendships may be more impacted by autonomy-supportive parenting than girls’ friendships when examined in Western cultures [24]. However, in China, girls are more often guided and educated to get involved in daily domestic tasks [48], and parents tend to enact stricter rules for girls their boys to protect them [49]. Accordingly, it can be speculated that, in contrast to the West, girls in China might be more susceptible than boys to the impact of autonomy-supportive parenting [48]. 

### 1.4. The Current Study

The primary aim of the current study was to examine bidirectional associations between maternal and paternal autonomy-supportive practices and peer preference in a sample of parents and older children (i.e., ages 9–12 years) in Mainland China. A longitudinal design was employed, spanning 18 months and including assessments at three time points. To date, little is known about the nature of the links among these constructs over time, particularly in non-Western cultures such as China. In keeping with previously established protocols [50], we employed a cross-lagged panel model to help to better understand how different contexts influenced one another in the development process. Drawing upon the extant literature, we postulated that: (1) parental autonomy-support would positively predict later peer preference; (2) peer preference would positively predict later parental autonomy-support; and (3) bidirectional associations between autonomy-supportive parenting and peer preference would be stronger for girls than boys.

## 2. Materials and Methods

### 2.1. Participants

Participants at Time 1 were *N* = 758 children in grades 4–6 (50.8% boys; *M*_age_ = 10.78 years, *SD* = 1.03) and their parents (*M*_age-father_ = 38.03 years, *SD* = 4.52 years; *M*_age-mother_ = 35.92 years, *SD* = 4.48). Children were recruited from four randomly selected public primary schools in Shanghai, P. R. China. Each class had approximately 30 students. Participants were from in families with low- to medium-SES backgrounds and were nearly all of Han nationality (the predominant nationality constituting 90% of the population in China). About 58% of mothers and 56% of fathers obtained a primary school or high school education, and 38% of mothers and 43 % of fathers received a college degree. Almost two-thirds (66.5%) of children were only children.

The follow-up data collections were conducted in the same schools one year later (Time 2), and again a year after that (Time 3). Data collection at each time point was conducted approximately during the middle of the school year (November and December). Overall, sample attrition resulted in a moderate decrease from Time 1 (*N =* 758) to Time 2 *N =* 672) and a larger decrease at Time 3 (*N =* 596) due to school transitions. Missing data rates for the study variables were low at Time 1 (ranging from 0 to 3%), increasing somewhat at Time 2 (11.3% to 13.6%), and again at Time 3 (ranging from 21.4% to 25.3%). 

The age of children at each time point was *M* = 10.73 years (*SD* = 1.22) at Time 1; *M* = 11.72 years (*SD* = 1.11) at Time 2; and *M* = 12.65 years (*SD* = 0.95) at Time 3. The result of Little’s MCAR test was non-significant, *χ*^2^(156) = 218.81, *p* > 0.05, indicating that the pattern of missingness was not systematic. Attrition analyses showed that there were no significant differences between participants who completed the data for all times points and those lost to attrition for all study variables except paternal autonomy-supportive parenting at Time 1, which was higher among those who discontinued participation (*M* = 3.03, *SD* = 1.17) as compared to those who remained (*M* = 2.84, *SD* = 1.07; *t* = 2.034, *p* = 0.042). We employed Full Information Maximum Likelihood (FIML) estimation to impute missing data.

### 2.2. Measures

#### 2.2.1. Autonomy-Supportive Parenting 

Children were asked to report how frequently their fathers and mothers exhibit autonomy-supportive practices using the Parenting Styles and Dimensions Questionnaire (PSDQ) [51]. The autonomy-support subscale is comprised of five items that reflect different aspects of autonomy- supportive behaviors, including acknowledgement of children’s feelings (e.g., “I am encouraged by my mother to express myself freely even when she disagrees with me”); encouragement of self-expression (e.g., “My mother allows me to express my ideas about family rules”); and providing explanations and reasons for rules (e.g., “My mother guide me make sense why some rules should be obeyed”). Items were rated on a 5-point Likert scale (from “never” to “always”. In the current sample, these five items loaded onto a single factor (all item loadings > 0.40) and demonstrated acceptable internal reliability across the three time points (α = 0.71–0.83). As a self-report measure, the PSDQ has previously been validated with Chinese children [33]. 

#### 2.2.2. Peer Preference

Children were asked to nominate up to three classmates whom they liked the most (positive nominations) and up to three whom they disliked the most (negative nominations). We summed and standardized nominations received from all children in each class. Following previously established protocols [52], cross-gender nominations were also permitted. As established by Coie et al. [53], and as previously reported in samples of Chinese children [54], an index of peer preference (i.e., indicator of the extent to which a child is liked by their classmates) was computed by subtracting children’s standardized negative nomination scores from their standardized positive nomination scores. Accordingly, children who received (relatively) more negative than positive nominations would obtain a peer preference score with a negative value (i.e., <0). In the current sample, peer preference scores ranged from −6 to 4. 

### 2.3. Procedure

The study was conducted in conformity with the Declaration of Helsinki and was approved by the ethical committee at East China Normal University. Before carrying out data collection, we obtained approval from participating elementary schools and written informed consent from parents and children. The data were collected by trained postgraduate students. Research assistants were responsible for introducing the aim of the study, explaining the confidentiality of participants’ responses, and making sure that students understood the data collection procedures. After completing the data collection, a gift was provided for every child as compensation for participation in the study. The first time point of data collection was in 2013. 

### 2.4. Data Analytical Plan

We conducted descriptive analyses using SPSS Statistic 22. Intra-class correlations (ICCs) computed at the classroom and school levels were all less than 0.02, indicating no notable of the nested design (students within classrooms within schools). A series of mixed repeated measures MANOVAs was utilized to test possible overall effects of Gender, Time, and their interactions. Given that we were interested in the between-level individual differences in the relations among maternal autonomy-supportive parenting, paternal autonomy-supporting parenting, and peer preference, cross-lagged analyses were conducted in *Mplus*, version 7.4 [55]. The robust maximum likelihood estimator (MLR) was used to account for potential issues caused by the presence of non-normal data. In order to examine potential gender differences in these effects, we conducted a multi-group analysis, comparing an unconstrained model (where all cross-lagged paths were allowed to vary across gender) with a constrained model (where cross-lagged paths were set to be equal across gender). Fit indices adopted to measure the absolute model fit included the comparative fit index (CFI), Tucker-Lewis index (TLI), root mean square error of approximation (RMSEA), standardized root mean square residual (SRMR), and *χ*^2^ test of significance. CFI and TLI values > 0.90, as well as RMSEA and SRMR values < 0.08 were considered as indicators of adequate model fit [56]. Because the *χ*^2^ test of significance is susceptible to sample size [56], we reported this index but did not use it as an indicator of absolute model fit. Since MLR estimation was adopted in our analyses, we also used the Satorra–Bentler Scaled *χ*^2^ difference test to compare the nested models.

## 3. Results

### 3.1. Preliminary Analyses

Descriptive statistics for study variables at each time point are presented in Table 1. Results from a series of mixed repeated measures MANOVAs indicated significant main effects of child Gender for maternal autonomy-supportive parenting, paternal autonomy-supportive parenting, and peer preference, *F* (1, 756) = 13.52, 9.71, and 39.39, respectively, *ps* < 0.01. Overall, girls reported higher scores than boys for both maternal and paternal autonomy-supportive parenting, as well as peer preference. There were no other significant main effects or interactions. 

Table 2 displays inter-correlations among all study variables. Among the results, all study variables demonstrated considerable stability across the three time points. Additionally, at each time point, maternal and paternal autonomy-supportive were highly correlated, and both were significantly and positively associated with peer preference.

### 3.2. Cross-Lagged Panel Analyses

To examine the reciprocal links between peer preference and parental autonomy-support, cross-lagged panel analysis was employed. At the same time, we controlled for the stability of each construct over time (i.e., focusing on *changes* in variables across time). The results showed that the model fit the data adequately, *χ*^2^(10) = 50.81, *p* < 0.001, CFI = 0.98, TLI = 0.93, RMSEA = 0.07, SRMR = 0.07 (see Figure 1). Results from follow up multi-group analyses indicated that there was no significant difference between the fit of the constrained and unconstrained models, Δ*χ*^2^(8) = 13.69, *p* = 0.09, ΔCFI = 0.003, ΔRMSEA = 0.007, ΔSRMR = 0.009, indicating that these reciprocal effects did not differ significantly by child gender. Higher peer preference predicted increases in both maternal and parental autonomy-supportive behaviors at subsequent time points, whereas parental autonomy-support did not significantly predict later changes in peer preference.

## 4. Discussion

Associations between autonomy-supportive parenting and aspects of children’s peer relationships have been well-documented in the literature [10,13]. However, less is known about how these constructs develop longitudinally and reciprocally, particularly in non-Western cultures. Most previous studies have also focused on effects driven by parents (i.e., from autonomy-support to child outcomes) [24,57]. As such, the dynamic and transactional processes linking parenting and peer relationships remains under-explored. Accordingly, the present study adopted a longitudinal design to examine the reciprocal relations between both maternal and paternal autonomy-support and children’s peer preference over time in the cultural context of mainland China. Overall, results revealed significant child-driven effects, with higher peer preference predicting an increase in later parental engagement in autonomy-supportive behaviors. In contrast, no significant predictive effects of parental autonomy- support on subsequent peer preference were found. In addition, the connections between autonomy-support granted by parents and peer preference appeared to be similar for boys and girls. 

### 4.1. Relations between Parental Autonomy-Support and Peer Preference 

We found concurrent positive associations between both maternal and paternal autonomy-support peer preference at each time point. These findings are consistent with previous results in samples of Western children [58,59], and bolster the notion that parental autonomy-support satisfies some of children’s basic psychological needs and ameliorates social adjustment. 

More novel and noteworthy are our findings related to the *reciprocal* associations between parental autonomy-support and children’s peer preference. In this regard, the pattern of results indicated significant effects from peers to parents, but not from parents to peers. Specifically, the cross-lagged model revealed that maternal and paternal autonomy-support did not significantly predict subsequent changes in peer preference. This aspect of the findings are consistent previous reports of non-significant effects for autonomy-support in studies conducted in Asian cultures [59]. In Western cultural contexts, the ability to behave independently is considered a pivotal indicator of child psychosocial maturity, and as such, parents’ efforts to encourage children’s autonomy are highly valued [60]. In contrast, achieving social harmony and appropriate behavioral control in interpersonal and childrearing contexts are more highly emphasized in Chinese culture [61]. As a result, there is a greater tendency for children to consider themselves as interconnected with their parents, and autonomy promoted by parents may be considered as a sign of a lack parental concern and caring [62,63]. From this point of view, the impact of parental autonomy-support on child social adjustment in China might be less pronounced than in the West. 

In contrast to the lack of parental effects, results from the current study provided strong evidence of child-driven effects. In line with our hypotheses, children’s peer preference positively predicted change in both maternal and paternal autonomy-support at later time points. Parents might consider adolescents who are well-liked by their peers as thriving and well-adjusted. Moreover, adolescents with better peer relationships are likely to be more socially competent, possess better self-regulation skills, and demonstrate fewer behavior problems [64,65]. Positive peer relationships also serve to reduce parent-adolescent conflicts [66]. Taken together, success in the peer realm might then encourage supportive behaviors by parents, providing their children with more choices and allowing them to act upon their own intentions and interests. In turn, this would help to further maintain positive relationships between adolescents and parents, which would likely continue to elicit more open and supportive parenting behaviors. The positive emotional feedback experienced by parents from their interactions with children would continue to invite autonomy-supportive behaviors, resulting in children experiencing more warmth, respect, and support from parents. 

### 4.2. Gender Effects

Overall, girls reported higher scores than boys for parental autonomy-support (both maternal and paternal) and peer preference. These results correspond with previous findings among both Western [62] and Chinese children [30,59]. Girls typically experience earlier maturation than boys, and they tend to demonstrate fewer problem behaviors than in boys [67]. In this regard, parents may consider girls as more capable of making their own decisions and thus provide more autonomy for daughters than sons. Overall, girls also typically possess superior perspective-taking abilities and higher levels of empathy than boys [68]. If girls are better at understanding others’ emotions and expressing their empathetic feelings, they would also be more likely to establish better interpersonal relationships and become more accepted by their peers than boys. 

Notwithstanding, it should also be noted that the links between parental autonomy-support and child peer preference did not differ significantly across gender. These results were not in line with our hypotheses, as we postulated that these links might be more pronounced among girls. One possible explanation for the absence of gender differences lies in the transition period experienced by children in our study. The current sample was comprised of children initially in grades 4–6, who transitioned from primary school to middle school across the two years of study. This represents a critical period that likely contributed to substantial changes in both boys’ and girls’ peer relationships. Adolescents may have left their friends at previous schools and had to establish new relationships with others in a novel environment, perhaps leading to greater instability in their reciprocal peer relationships for both genders [69]. In other words, boys and girls experience similar levels of friendship instability in early adolescence. Consequently, parents may respond to children’s changing social relationships similarly across both genders, supporting their autonomous selection and formation of peer relationships in the new school environment. Of note, there is also at least some evidence to suggest that child effects may become even more prominent in adolescence, when the parent-child relationship becomes more egalitarian (which appears to apply to both boys and girls) [70]. Parents may thus be more impacted by peer-related factors across adolescence as their relationships with children become more symmetric. The lack of previous gender differences in these factors may have served to attenuate the proposed gender effects.

### 4.3. Limitations and Future Directions 

The findings from this study contribute to a better understanding of the dynamic developmental processes that link parental autonomy-support and child peer relations, in the unique context of a non-Western culture. Despite these novel insights, some limitations should be considered when interpreting the results and with an eye toward future directions. First, participants of this study were recruited from Shanghai, one of the most highly developed urban centers in China. Substantial differences still exist in both social and economic domains across different regions in China, and as such, our results may not be generalizable to other parts of China (e.g., rural areas) [71]. The present study also considered peer preference as the sole indicator of children’s social adjustment. Future research should consider other group-level or dyadic-level peer experiences that might also be associated with autonomy-support, such as friendship quality [53]. Also, participants in this study were followed from late childhood to early adolescence. Given previous evidence that the benefits of parental autonomy-support continue to increase into later adolescence [72], it would be of interest to expand the investigation into later development stages. 

In terms of other methodological issues, child self-reports were utilized to assess perceptions of parental-autonomy-support, which may differ from parent-rated or observed parenting practices. The inclusion of self-reports (rather than third-party assessments) was deemed most appropriate for this study because children’s perceptions of being supported by parents are personal and subjective experiences [73]. Nevertheless, further investigations might also include additional sources of assessment to provide a more thorough understanding of the dynamic associations between parenting and children’s social adjustment from multiple perspectives. Finally, from an analysis perspective, there are alternative statistical approaches (e.g., Random-Intercept Cross-Lagged Panel Model) that could be used to further investigate the nature of the relations among these constructs over time.

Despite these limitations, the current study shed new light on the potentially intricate processes linking parenting and children’s peer relationships in Chinese culture. Our findings also have some potentially important practical implications. For example, if parents are made more explicitly aware of the impact that children’s peer experiences appear to exert over their parenting behaviors, they may be able to better adjust their own adaptive responses. In addition, our findings emphasize the positive effects of an autonomous family environment for both boys’ and girls’ social development, and parents are encouraged to afford their children more autonomy to further promote positive peer experiences.

## Figures and Tables

**Figure 1 children-08-00508-f001:**
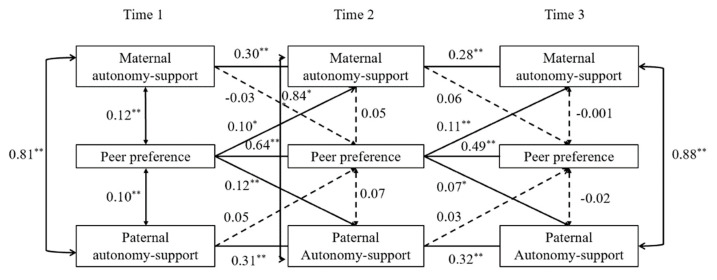
Cross-lagged model of maternal and paternal autonomy-support and peer preference. Note: Values represent standardized coefficients. Dashed lines represent non-significant paths. * *p* < 0.05, ** *p* < 0.01.

**Table 1 children-08-00508-t001:** Descriptive statistics for all study variables.

	Boys		Girls	
	*M*	*SD*	*M*	*SD*
Maternal autonomy-supportive parenting				
Time 1	2.87	1.04	3.01	1.11
Time 2	2.79	1.17	3.01	1.21
Time 3	2.92	1.03	3.08	1.21
Paternal autonomy-supportive parenting				
Time 1	2.84	1.06	2.92	1.13
Time 2	2.72	1.28	2.92	1.29
Time 3	2.81	1.04	3.00	1.20
Peer preference				
Time 1	−0.22	1.71	0.23	1.39
Time 2	−0.25	1.68	0.22	1.51
Time 3	−0.27	1.92	0.11	1.61

Note: M = mean value; SD = standard deviation.

**Table 2 children-08-00508-t002:** Inter-correlations among study variables.

	1	2	3	4	5	6	7	8	9
1. MA (T1)	-								
2. MA (T2)	0.37 **	-							
3. MA (T3)	0.23 **	0.26 **	-						
4. PA (T1)	0.79 **	0.34 **	0.35 **	-					
5. PA (T2)	0.35 **	0.84 **	0.25 **	0.38 **	-				
6. PA (T3)	0.35 **	0.48 **	0.56 **	0.35 **	0.49 **	-			
7. Peer Preference (T1)	0.13 **	0.17 **	0.09 *	0.10 **	0.16 **	0.17 **	-		
8. Peer Preference (T2)	0.08 *	0.10 **	0.10 *	0.08 *	0.12 **	0.08 *	0.58 **	-	
9. Peer Preference (T3)	0.13 **	0.12 **	0.07 *	0.14 **	0.13 **	0.07 *	0.48 **	0.60 **	-

Note: MA = maternal autonomy-support; PA = paternal autonomy-support. * *p* < 0.05, ** *p* < 0.01.

## Data Availability

The data presented in this study are available on request from the corresponding author. The data are not publicly available due to ethical principles.

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
