# Peer review of "Examining Reciprocal Links between Parental Autonomy-Support and Children’s Peer Preference in Mainland China"

_children, 2021, doi:10.3390/children8060508_

Round 1
Reviewer 1 Report
The definition of autonomy support in the first sentence needs some refinement. First, providing guidance is better framed within structure, based upon SDT (see Grolnick, 2016). Also, highlighting interesting or meaningful features of tasks or treating others well is also a core part of both parental autonomy support.
The mention of mixed results among Asian samples should be qualified by mentioning that autonomy support does promote psychological benefits in east Asia, however, the effects of controlling parenting may be mediated by sense of collectivism (Soenens et al. 2018).
Please share more examples of the parental autonomy support items so that readers can see how each aspect of autonomy support is covered. Also, are there any items that tap into guidance? If so, these would best be removed from the scale, unless you want to add a separate measure of structure.
Social nominations are quite different than positive peer relationships. The abstract and title should be re-written to reflect this. One could be controversial and still have very positive relationships with those that like her. Also, the intro should include more on social nominations (accepted, reject, controversial, etc.).
References
Grolnick, W. S. (2016). Parental involvement and children's academic motivation and achievement. In W. C. Liu, J. C. K. Wang, & R. M. Ryan (Eds.), Building autonomous learners: Perspectives from research and practice using self-determination theory (pp. 169-183). Springer.
Soenens, B., Park, S. Y., Mabbe, E., Vansteenkiste, M., Chen, B., Van Petegem, S., & Brenning, K. (2018). The moderating role of vertical collectivism in South-Korean adolescents’ perceptions of and responses to autonomy-supportive and controlling parenting. Frontiers in psychology, 9, 1080.
Reviewer 2 Report
The current manuscript examined the reciprocal relations between parental autonomy support and children’s peer preferences in a sample of 758 children from Shanghai who were followed over a three-year period. Results indicated that peer preferences were predictive of later parental autonomy support but not vice versa. Strengths of the study include the clear rationale set out in the introduction, the utilisation of a large longitudinal sample with a focus on an Asian population and the inclusion of gender invariance tests. While the manuscript has a number of strengths, I have some major concerns, particularly relating to the statistical analysis. These are summarized below:
- My main concern relates to the use of a cross-lagged panel model. It has recently been established that CLPMs suffer from a major limitation, that is they conflate within- and between-person/family effects which, in the context of the current study, is problematic as, arguably, the effect of peer preference on parental autonomy support refers to a within-person/family effect. There are a number of alternatives to the CLPM that appropriately disaggregate within- and between-person effects, for instance, the random-intercept CLPM or the autoregressive latent trajectory model with structured residuals. It is not clear to me why one of these models wasn’t chosen as the data should allow for these analyses. See the following papers for in-depth discussions of the within/between-person issues and alternative models:
- Hamaker, E. L., Kuiper, R. M., & Grasman, R. P. P. P. (2015). A critique of the cross-lagged panel model. Psychological Methods, 20(1), 102–116. https://doi.org/10.1037/a0038889
- Berry, D., & Willoughby, M. T. (2017). On the Practical Interpretability of Cross-Lagged Panel Models: Rethinking a Developmental Workhorse. In Child Development (Vol. 88, Issue 4, pp. 1186–1206). Blackwell Publishing Inc. https://doi.org/10.1111/cdev.12660
- Curran, P. J., & Hancock, G. R. (2021). The Challenge of Modeling Co‐Developmental Processes over Time. Child Development Perspectives, cdep.12401. https://doi.org/10.1111/cdep.12401
- Curran, P. J., Howard, A. L., Bainter, S. A., Lane, S. T., & McGinley, J. S. (2014). The separation of between-person and within-person components of individual change over time: A latent curve model with structured residuals. Journal of Consulting and Clinical Psychology, 82(5), 879–894. https://doi.org/10.1037/a0035297
- A second issue relating to the statistical analysis is that from my reading of the analysis section, children from the same school were treated as independent observations. However, children clustered within the same higher-level structure tend to be more similar to each other as they share risk factors (e.g. a non-supportive or a supportive teacher) and should therefore not be treated as independent observations as seems to be the case in the current statistical analysis. The authors might want to consider accounting for the multilevel structure of the data in their analysis or at least present some data to reassure the reader that the observations were indeed mostly independent (e.g. intraclass correlation coefficients).
- It is not quite clear to me what results were used as a basis for discussing concurrent effects. From the CLPM figure, it looks as if residual covariances were included for the first time point but not for the second and third. Why? Please add more details on the structure of the model to the analysis section. I assume that interpretations of concurrent effects are based on the correlations presented in Table 2. However, the interpretation of the data’s correlational structure itself does not allow for a conclusion on directional effects. Please add some more information on how concurrent relations were assessed (consider adding within-time point covariances for each time-point) and avoid causal language when discussing their results. In particular, please consider rewording the final sentence of the strengths and limitation section which currently implies a causal relation that is not supported by the results (Page 9, line 362-365).
Minor Points:
- In the introduction section on parental effects (Page 2, line 53-55), you briefly discuss Updegraff et al.’s study which “found that paternal autonomy-support in open communications with sons was positively related to the level of intimacy in adolescent friendship”. This seemed to imply that, in this particular study, boys rather than girls, were more likely to be positively affected by paternal autonomy-support. For some reason, this gender difference is not discussed in the section on the role of gender which concludes that girls are more likely than boys to be susceptible to parental impact. Please clarify this conflicting evidence.
- Please add information on children’s ages to the current study section (Page 3).
- Could you add some information on the attrition analysis to supplementary materials (Page 4)?
- In the discussion, you state that “the present study adopted a longitudinal design with multiple informants”. I think the use of “multiple informants” here is misleading as it implies that you had different raters which you didn't as you rightly state in the limitations section. Did you mean that you included both maternal and paternal autonomy support? Please clarify (Page 7, line 257).
- Please clarify that the result on child effects being the same for boys and girls was not in line with your hypotheses (Page 7, line 289).
Author Response
Response to Reviewer 2 Comments
- My main concern relates to the use of a cross-lagged panel model. It has recently been established that CLPMs suffer from a major limitation, that is they conflate within- and between-person/family effects which, in the context of the current study, is problematic as, arguably, the effect of peer preference on parental autonomy support refers to a within-person/family effect. There are a number of alternatives to the CLPM that appropriately disaggregate within- and between-person effects, for instance, the random-intercept CLPM or the autoregressive latent trajectory model with structured residuals. It is not clear to me why one of these models wasn’t chosen as the data should allow for these analyses. See the following papers for in-depth discussions of the within/between-person issues and alternative models:
- Hamaker, E. L., Kuiper, R. M., & Grasman, R. P. P. P. (2015). A critique of the cross-lagged panel model. Psychological Methods,20(1), 102–116. https://doi.org/10.1037/a0038889
- Berry, D., & Willoughby, M. T. (2017). On the Practical Interpretability of Cross-Lagged Panel Models: Rethinking a Developmental Workhorse. In Child Development(Vol. 88, Issue 4, pp. 1186–1206). Blackwell Publishing Inc. https://doi.org/10.1111/cdev.12660
- Curran, P. J., & Hancock, G. R. (2021). The Challenge of Modeling Co‐Developmental Processes over Time. Child Development Perspectives, cdep.12401. https://doi.org/10.1111/cdep.12401
- Curran, P. J., Howard, A. L., Bainter, S. A., Lane, S. T., & McGinley, J. S. (2014). The separation of between-person and within-person components of individual change over time: A latent curve model with structured residuals. Journal of Consulting and Clinical Psychology, 82(5), 879–894. https://doi.org/10.1037/a0035297.
Response: Thank you to this Reviewer for raising this interesting point of discussion. We are aware of the Random–Intercept Cross-Lagged Panel Model as an alternative statistical approach to assess causal processes using longitudinal data. For the purposes of the present study - there are several reasons why we used the Cross-Lagged Panel Model and we believe that it is the more appropriate analysis.
(1) From a conceptual perspective, we are interested in the between-level individual differences in the relations among maternal autonomy support parenting, paternal autonomy support parenting and peer preference. Therefore, our research focused on individual differences (interindividual variability) rather than within-individual changes (intraindividual variability). Selig and Little (2012) have suggested that although the parameters of the panel model are affected by intraindividual change, the parameters of the panel model are not sensitive to the type of individual-level change. For example, they stated that “The panel model is especially useful for identifying the relations between variables across time” (p. 265), and “Panel model are a poor choice only when the goal is to show the functional form of systematic (mean-level) change in a variable over time or if the theoretical emphasis is on intraindividual change” (p. 267). On the other hand, Hamaker’s study (2015) only indicated that the estimates from standard cross-lagged models might have a limited validity only when it comes to understanding the within-person effects. Further, according to Keijsers (2016), if the research is aimed at understanding dynamics of people rather than populations, it may be crucial to examine associations at the level where causality takes place: the within-person level, otherwise the panel model is appropriate.
(2) When we calculated intra-class correlations across waves (measures within persons), for maternal autonomy-support parenting, the ICC was .9957. This indicates that 99.57% of the variance in the three measures of maternal autonomy-support parenting is explained by differences between children, and the remainder (0.43%) was explained by within-person fluctuations. Likewise, the ICCs for paternal autonomy-support parenting and peer preference were 99.87% and 99.93%, respectively. Hence, for each of the variables in this study, only a very small portion of the variance is due to fluctuations over time.
(3) Keijsers (2016) found that Random Intercept Cross-Lagged Panel Models with one-year intervals was less optimal in detecting causal effects over time, even when controlling for stable between-person differences.
Accordingly, we have retained our original approach to statistical analyses. Notwithstanding, we now acknowledge in the Limitations and Future Directions section that there are alternative statistical approaches such as the Random-Intercept Cross-Lagged Panel Model that could be used to investigate the nature of the relations among these constructs and that could be explored in future research (see p. 9, lines 378-380).
References:
Hamaker, E. L., Kuipers, R. M., & Grasman, R. P. (2015). A critique of the cross-lagged panel model. Psychological Methods, 20, 102-116.
Keijsers, L. (2016). Parental monitoring and adolescent problem behaviors: How much do we really know?. International Journal of Behavioral Development, 40, 271-281.
Selig, J.P., & Little, T. D. (2012). Autoregressive and cross-lagged panel analysis for longitudinal data. In B. Laursen, T. D. Little, & N. A., Card (Eds), Handbook of developmental research methods (pp. 265-278). New York, NY: The Guilford Press.
- A second issue relating to the statistical analysis is that from my reading of the analysis section, children from the same school were treated as independent observations. However, children clustered within the same higher-level structure tend to be more similar to each other as they share risk factors (e.g. a non-supportive or a supportive teacher) and should therefore not be treated as independent observations as seems to be the case in the current statistical analysis. The authors might want to consider accounting for the multilevel structure of the data in their analysis or at least present some data to reassure the reader that the observations were indeed mostly independent (e.g. intraclass correlation coefficients).
Response: Thanks for your suggestions. We calculated ICCs (students within classroom) and the results indicated that the nested structure was not an issue of concern (all ICCs were less than 2%) and we have added this information to the manuscript (p. 6, lines 246-248).
- It is not quite clear to me what results were used as a basis for discussing concurrent effects. From the CLPM figure, it looks as if residual covariances were included for the first time point but not for the second and third. Why? Please add more details on the structure of the model to the analysis section. I assume that interpretations of concurrent effects are based on the correlations presented in Table 2. However, the interpretation of the data’s correlational structure itself does not allow for a conclusion on directional effects. Please add some more information on how concurrent relations were assessed (consider adding within-time point covariances for each time-point) and avoid causal language when discussing their results. In particular, please consider rewording the final sentence of the strengths and limitation section which currently implies a causal relation that is not supported by the results (Page 9, line 362-365).
Response: We have added the concurrent relations in the model (see Figure 1) and avoided causal language when discussing the results. We also reworded the final sentence of the strengths and limitation section.
Minor Points:
- In the introduction section on parental effects (Page 2, line 53-55), you briefly discuss Updegraff et al.’s study which “found that paternal autonomy-support in open communications with sons was positively related to the level of intimacy in adolescent friendship”. This seemed to imply that, in this particular study, boys rather than girls, were more likely to be positively affected by paternal autonomy-support. For some reason, this gender difference is not discussed in the section on the role of gender which concludes that girls are more likely than boys to be susceptible to parental impact. Please clarify this conflicting evidence.
Response: We agree with your ideas about the finding from the Updegraff et al’s study stressed that boys rather than girls were more likely to be positively affected by paternal autonomy-support. However, this gender difference occurred in the West and may not be as applicable in China. We now mention this in the Introduction (see p.3, lines 134-135).
- Please add information on children’s ages to the current study section (Page 3).
Response: We have added the information in the manuscript (see p. 3, line 143).
- Could you add some information on the attrition analysis to supplementary materials (Page 4)?
Response: Thanks for your comment. The attrition rates of maternal autonomy-support parenting over time are 10.8% and 13.4%; of paternal autonomy-support parenting over time are 12% and 13.9%; of peer preference over time are 11.3% and 11.3%. Attrition analyses showed that there were no significant difference between participants who completed the data for all times points and those lost to attrition on children sex, χ2(1)=3.557, p>.05. Additionally, only paternal autonomy-support parenting at Time1 was significantly different between dropped children and not dropped (t = 2.034, p = .042; Mdropped = 3.03, SD=1.17; Mnot-dropped = 2.84, SD=1.07). We have added the information in the manuscript (see p. 5, lines 198-202).
- In the discussion, you state that “the present study adopted a longitudinal design with multiple informants”. I think the use of “multiple informants” here is misleading as it implies that you had different raters which you didn't as you rightly state in the limitations section. Did you mean that you included both maternal and paternal autonomy support? Please clarify (Page 7, line 257).
Response: We acknowledge that the use of “multiple informants” here is misleading. As a result, we changed the specific description of this sentence (see p. 7, lines 285-288).
- Please clarify that the result on child effects being the same for boys and girls was not in line with your hypotheses (Page 7, line 289).
Response:We now discuss the results for parent effects and child effects being the same for boys and girls separately in the Discussion (see p. 8).
Round 2
Reviewer 2 Report
Many thanks for the opportunity to re-review this manuscript. I am happy as to how the authors have modified the manuscript according to my former comments. Just a few minor things:
- I think it would be worth explicitly highlighting in the Introduction or Methods section that you are primarily interested in between-level individual differences
- Line 15: Mage = 11.72 years, SD = 1.11 at Wave 1; Should this be Wave 2?
Author Response
Response to Reviewer 2 Comments
1. I think it would be worth explicitly highlighting in the Introduction or Methods section that you are primarily interested in between-level individual differences.
Response: Thanks for your comments. We have added some information in the section of Method, please see p. 6, line 251-253.
2. Line 15: Mage = 11.72 years, SD = 1.11 at Wave 1; Should this be Wave 2?
Response: We have revised the mistake, please see line 15.